# Restrictive Use of Empirical Antibiotics Is Associated with Improved Short Term Outcomes in Very Low Birth Weight Infants: A Single Center, Retrospective Cohort Study from China

**DOI:** 10.3390/antibiotics12040741

**Published:** 2023-04-12

**Authors:** Meiyan Chu, Jing Lin, Mingjie Wang, Zhengchang Liao, Chuanding Cao, Ming Hu, Ying Ding, Yang Liu, Shaojie Yue

**Affiliations:** 1Department of Neonatology, Xiangya Hospital of Central South University, Changsha 410008, China; 2Department of Pediatrics, Icahn School of Medicine at Mount Sinai, New York, NY 10029, USA; 3Xiangya School of Public Health, Central South University, Changsha 410078, China

**Keywords:** antibiotics, antibiotic stewardship, very low birth weight infant, neonatal sepsis, China

## Abstract

Antibiotics are essential for treating neonatal sepsis, but abuse or inappropriate use of antibiotics have harmful adverse effects. The inappropriate use of antibiotics has led to the significant increase in bacterial antimicrobial resistance in the neonatal intensive care unit (NICU). The aim of this study was to retrospectively analyze the changes in antibiotic usages in a NICU after the implementation of an antibiotic stewardship program and to determine the impact of this implementation on the short-term clinical outcomes of very low birth weight (VLBW) infants. The antibiotic stewardship program was initiated in the NICU in early 2015. For analysis, all eligible VLBW infants born from 1 January 2014 to 31 December 2016 were enrolled, and we classified the year 2014 as pre-stewardship, 2015 as during stewardship, and 2016 as post-stewardship. A total of 249 VLBW infants, including 96 cases in the 2014 group, 77 cases in the 2015 group, and 76 cases in the 2016 group, were included for final analysis. Empirical antibiotics were used in over 90% of VLBW infants in all three groups during their NICU stay. Over the 3-year period, the duration of an initial antibiotic course was significantly reduced. The proportion of patients receiving an initial antibiotic course for ≤3 days gradually increased (2.1% vs. 9.1% vs. 38.2%, *p* < 0.001), while the proportion of babies treated with an initial antibiotic course >7 days significantly decreased (95.8% vs. 79.2% vs. 39.5%, *p* < 0.001). The total days of antibiotic usage during the entire NICU stay also showed a significant reduction (27.0 vs. 21.0 vs. 10.0, *p* < 0.001). After adjusting for confounders, the reduction in antibiotic usage was associated with decreased odds of having an adverse composite short-term outcome (aOR = 5.148, 95% CI: 1.598 to 16.583, *p* = 0.006). To assess the continuity of antibiotic stewardship in the NICU, data from 2021 were also analyzed and compared to 2016. The median duration of an initial antibiotic course further decreased from 5.0 days in 2016 to 4.0 days in 2021 (*p* < 0.001). The proportion of an initial antibiotic course in which antibiotics were used for ≤3 days increased (38.2% vs. 56.7%, *p* = 0.022). Total antibiotic usage days during the entire NICU stay also decreased from 10.0 days in 2016 to 7.0 days in 2021 (*p* = 0.010). The finding of this study strongly suggests that restricting antibiotic use in VLBW infants is beneficial and can be achieved safely and effectively in China.

## 1. Introduction

Neonatal sepsis remains one of the leading causes for neonatal mortality and morbidity. Antibiotics are essential for treating neonatal sepsis, but abuse or inappropriate use of antibiotics have harmful adverse effects. A number of studies have suggested that the unnecessary early use of antibiotics increases the incidence in premature infants of clinically adverse outcomes such as late-onset sepsis (LOS), necrotizing enterocolitis (NEC), and severe intraventricular hemorrhage (IVH) [1,2]. Other studies have found an association between early prolonged antibiotic exposure and the risk of developing asthma and obesity in childhood [3,4].

More importantly, the inappropriate use of antibiotics has led to a significant increase in bacterial antimicrobial resistance and the spread of bacterial antibiotic-resistant genes. As a result, neonatal multidrug-resistant (MDR) bacterial infections are now a global challenge [5]. By merging data from five developing nations (India, Nigeria, the Democratic Republic of the Congo, Pakistan, and China), Laxminaraya et al. [6] showed that MDR bacterial infections account for more than 50% of fatalities due to neonatal sepsis globally. Li JY, et al. [7] recently analyzed data collected from the Chinese literature and revealed that more than 60% of isolated Staphylococcus aureus were methicillin-resistant and that more than 50% of gram-negative pathogens, including *Escherichia* spp. and *Klebsiella*, were resistant to the widely used third-generation cephalosporins. The increase in MDR bacterial infections in newborns due to antibiotic abuse has made treatment more difficult and economically burdensome for developing countries [5]. This has now become one of the major global health issues, which the World Health Organization (WHO) is attempting to address [8].

Wang MJ et al. recently reported survey data about the abuse of antibiotics in 24 neonatal intensive care units (NICU) in Hunan Province, China. They found that in very low birth weight (VLBW) infants (newborns with birth weight less than 1500 g), on average, antibiotics were administered for more than half of the duration of their hospital stay [9]. Recognizing the problem of antibiotic abuse in VLBW infants, an antibiotic stewardship program was first introduced in 2015 to the neonatologists and staff members in the NICU, a level 4 perinatal center located in the capital of Hunan Province. The specific guidelines for the diagnosis and treatment of early onset sepsis (EOS) in VLBW infants were induced and implemented. The specific guidelines used for EOS were published in China recently [10]. The implemented guidelines for late onset sepsis (LOS) were consistent with the recommendations about the diagnosis and treatment of neonatal sepsis which were published jointly by the Neonatology Academic Committee of the Chinese Pediatric Society and the Infection Committee of the Chinese Neonatologist Association [11,12,13]. The aim of this study was to retrospectively analyze the changes in antibiotic usages in the NICU of Xiangya Hospital of Central South University after the implementation of an antibiotic stewardship program, and to determine the impact of this implementation on short-term clinical outcomes of VLBW infants.

## 2. Results

### 2.1. General Characteristics of the Study Subjects

A total of 351 VLBW infants were admitted during the study period. The flow diagram of enrollment is presented in Figure 1. The excluded subjects were 40 cases who were admitted after 24 h of birth, 15 cases with incomplete clinical data, 8 cases with severe congenital anomalies, 6 cases with suspected or confirmed genetic metabolic disorders, 6 cases with EOS confirmed by blood or cerebrospinal fluid culture, and 27 cases who died within the first week of life. A total of 249 VLBW infants, including 96 in the 2014 group, 77 in the 2015 group, and 76 in the 2016 group, were included in the final analysis (Figure 1).

The differences among the three groups in birth weight, gestational age, small for gestational age (SGA), in vitro fertilization (IVF), premature rapture of membrane (PROM), cesarean delivery, invasive mechanical ventilation for the first 3 days after birth, and antepartum antibiotics usage were not statistically significant (all *p* > 0.05). The proportion of Apgar scores ≤ 7 at 5 min was higher in the 2016 group than in the 2014 group (13.2% vs. 4.2%, *p* = 0.042) (Table 1).

### 2.2. Postnatal Empirical Antibiotic Usage Changes and the Short-Term Outcomes

Empirical antibiotics were used in over 90% of VLBW infants in all three groups during their NICU stay. The most common first line antibiotics used for empirical treatment of suspected EOS in our unit were ampicillin/sulbactam, amoxicillin/clavulanic acid, or ampicillin plus one third generation cephalosporin. Over the 3-year period, with antibiotic stewardship being introduced in 2015, the duration of the initial antibiotic course was reduced significantly. The median duration of the initial antibiotic course decreased from 25.0 days in 2014 to 13.5 days in 2015 and then further decreased to 5.0 days in 2016 (*p* < 0.001). The proportion of the initial antibiotic courses in which antibiotics were used for ≤3 days gradually increased (2.1% vs. 9.1% vs. 38.2%, *p* < 0.001), while the proportion of initial antibiotic courses >7 days significantly decreased (95.8% vs. 79.2% vs. 39.5%, *p* < 0.001). Total antibiotic usage days during the entire NICU stay also showed a significant reduction, with the median total antibiotic usage days decreasing from 27.0 days in 2014 to 21.0 days in 2015 and then to 10.0 days in 2016 (*p* < 0.001). The antibiotic use rate (AUR) also declined significantly over the period of the study with the median of 64% in 2014 decreasing to 39% in 2015 and then further to 12% in 2016 (*p* < 0.001). These findings illustrate that the duration of antibiotic use in this NICU was significantly reduced after the initiation and implementation of the antibiotic stewardship program in 2015 (Table 2).

When comparing the short-term clinical outcomes among the three groups, no statistically significant differences were found between mortality, NEC, LOS, ≥3 stage retinopathy of prematurity (ROP), and severe bronchopulmonary dysplasia (BPD), (all *p* > 0.05). The difference in the days of hospitalization among the three groups was also not statistically significant (*p* > 0.05). However, the differences among the three groups were statistically significant in the incidence of severe IVH, with a lower incidence of severe IVH seen in 2016 than that in 2014 (7.9% vs. 25.0%, *p* = 0.013). There were significant reductions in the days to full enteral feedings over the 3 years. The median days to full enteral feeding in 2016 was 17.0 days as compared to 21.5 days in 2015, and 26.0 days in 2014 (*p* < 0.001). These data strongly suggest that reducing the antibiotic usage days over the 3-year period in VLBW infants was not associated with an increased incidence of worse short-term adverse clinical outcomes. If anything, it is associated with improved short-term outcomes in VLBW infants (Table 2).

### 2.3. Logistic Regression Analysis to Determine the Risk Factors

To further determine the association between antibiotic use and short-term adverse clinical outcomes, the presence of any of the following conditions are defined as composite adverse outcomes: death, NEC, LOS, severe IVH, severe ROP, and severe BPD. Using logistic regression analysis, an association was identified between the composite adverse outcome with both the number of days of early antibiotic use and the AUR. There was an increased risk of clinical composite adverse outcome in the >7 days group when compared to the ≤3 days group (*p* = 0.001), and an approximately 18% increase in risk of clinical composite adverse outcomes for every 10% increase in AUR (OR = 1.181, *p* < 0.001).

In a multivariate regression analysis model which corrected for confounders such as maternal age, IVF, antenatal steroids, PROM, cesarean delivery, multiple, male gender, gestational age, birth weight, Apgar score ≤ 7 at 5 min, endotracheal intubation in the delivery room, surfactant administration, and invasive mechanical ventilation for the first 3 days after birth, the risk of composite adverse clinical outcomes in VLBW infants was found to have increased by approximately 17% for every 10% increase in AUR (aOR = 1.175, *p* < 0.01). After adjusting for confounders, the risk of having a composite adverse outcome in infants who received more than 7 days of antibiotics as the initial antibiotic course was 5.1-fold (aOR = 5.148, 95% CI: 1.598 to 16.583, *p* = 0.006) higher than those received 0–3 days as the initial antibiotic course. Since there was a significant reduction in antibiotic use in 2016, the adjusted OR for composite adverse outcome in 2016 was also significantly reduced (*p* < 0.001) when using that of 2014 as a baseline reference (Table 3).

### 2.4. The Continued Effect of the Antibiotic Usage Strategy in 2021

VLBW infants in the 2021 group have greater gestational age, higher birth weight, lower proportion of males and surfactant usage, and a higher proportion of cesarean delivery compared to 2016 (all *p* < 0.05) (Table 4). Compared to 2016, early antibiotic usage rate in the 2021 group further decreased (92.1% vs. 66.7%, *p* < 0.001). The median duration of the initial antibiotic course decreased from 5.0 days in 2016 to 4.0 days in 2021 (*p* < 0.001). The proportion of the initial antibiotic courses in which antibiotics were used for ≤3 days increased (38.2% vs. 56.7%, *p* = 0.022), while the proportion of initial antibiotic courses > 7 days significantly decreased (39.5% vs. 21.1%, *p* = 0.022). Total antibiotic usage days during the entire NICU stay also decreased from 10.0 days in 2016 to 7.0 days in 2021 (*p* = 0.010). The AUR also declined significantly over the period of the study with the median of 12% in 2016 decreasing to 7% in 2021 (*p* < 0.001).

There were no statistically significant differences between mortality, NEC, IVH, LOS, severe ROP, and severe BPD between the 2016 and 2021 groups (all *p* > 0.05). The difference in the days of hospitalization among the two groups was not statistically significant (*p* > 0.05). However, the median of days to full enteral feedings in the 2021 groups statistically increased when compared to 2016 (24.0 vs. 17.0, *p* < 0.001). The results suggest that the antibiotic use strategy was maintained in 2021, and that antibiotic usage was substantially reduced (Table 5).

## 3. Discussion

In the current study, the authors demonstrate that the implementation of an antibiotic stewardship program in the NICU can significantly reduce the use of antibiotics in VLBW infants during the entire hospitalization. Over the 3-year period, the proportion of initial antibiotic courses lasting ≤3 days increased, and the proportion of courses >7 days decreased. The proportion of initial antibiotic courses lasting ≤3 days further increased, and the proportion of courses >7 days also decreased in 2021. The reduction in antibiotic usage in VLBW infants who had a negative blood culture on admission was associated with a decreased incidence of composite short-term adverse outcome as defined by the presence of any of the listed conditions (death, NEC, LOS, severe IVH, severe ROP, and severe BPD). Multivariate logistic regression analysis indicates that the risk of composite short-term adverse outcome in VLBW infants increases by approximately 17% for every 10% increase in AUR. After the implementation of an antibiotic stewardship program in 2015 in the NICU, the outcomes of the VLBW infants cared for in the NICU in 2016 were indeed improved. This study illustrates that by implementing an antibiotic stewardship program, the antibiotic abuse in VLBW infants in developing countries such as China can be safely and effectively reduced without increasing the clinical complications, the possibility of which concerned some care providers. The study also demonstrates that the adjustment to the antibiotic stewardship program is sustainable, confirming its safety. In fact, better outcomes may be achieved with more restrictive use of antibiotics in VLBW infants.

Premature infants, especially those with VLBW, are at high risk of developing bacterial infections or sepsis [14]. Furthermore, VLBW infants with immature organ systems often present with non-specific symptoms such as respiratory distress, temperature instability, feeding intolerance, and poor response. All symptoms related to prematurity are difficult to distinguish from the early clinical manifestations of sepsis. Therefore, it is very common for VLBW infants to receive some degree of empirical antibiotic treatment during their NICU stay. As such, most enrolled subjects were exposed to empirical antibiotics. Clearly, how to effectively and safely reduce the use of antibiotics in VLBW infants is a challenge. The American Academy of Pediatrics (AAP) has recently issued guidelines for the management of neonates born at ≤34 6/7 weeks gestation with suspected or proven EOS. They recommend that those premature infants born to mothers with lower risk factors for EOS may not need empirical antibiotics at admission into NICU [15]. Indeed, a retrospective analysis by Morales-Betancourt [16] et al. recently demonstrated that a reduction in antibiotic use in VLBW infants could be achieved through quality improvement in a NICU in Spain. By implementing an antibiotic stewardship program and a surveillance system, the proportion of VLBW infants administered early antibiotics and infants treated with antibiotics for longer than 3 days decreased significantly. Therefore, both this study and others indicate that the incidence of empirical antibiotic use in VLBW infants can be safely and effectively reduced via implementation of an antibiotic stewardship and quality improvement project.

It is recommended that empirical antibiotics may be discontinued if the blood culture remains negative after 48 h and the child’s clinical presentation does not indicate ongoing infection [15]. This suggests the importance of blood cultures as a criterion for the diagnosis of bacterial sepsis and also as a guide for the duration of antibiotic therapy in neonates [17]. Given that the majority of blood cultures of neonates with sepsis have a bacterial blood colony count of ≤10 CFU/mL, a portion of these patients may have false negative blood culture results if less than 1.0 mL of blood is drawn for culture [18]. For this reason, the diagnosis of culture negative sepsis is often made in neonates with clinical presentations consistent with sepsis and abnormal non-specific laboratory inflammatory indicators such as C-reactive protein (CRP), procalcitonin (PCT), and complete blood cell (CBC) counts. This is the main justification for premature infants being treated with antibiotics for prolonged periods without a positive blood culture. Data from this study showed that in 2014, 95.8% of the VLBW infants had their initial antibiotic course for >7 days. This is consistent with the authors’ previous report that in VLBW infants the total antibiotic use accounted for more than half of their entire hospital stay (mean AUR > 50%) in 24 NICUs of Hunan Province [9]. Similarly, Hou SS et al. [19] reported an AUR of 56% in VLBW infants from 24 NICUs of the Northern China Neonatal Network. In the United States, Flannery et al. [20] collected data from 297 NICUs and found that 78.6% of VLBW infants had early antibiotic initiation. Among infants who survived and remained in the NICU for more than 5 days, 26.5% of VLBW infants received prolonged antibiotic therapy defined as starting within 3 days of birth and continuing for >5 days. This suggests that the reduction in unnecessary antibiotic use in VLBW infants is a challenge in both developing and developed countries; although, the situation in China is much worse.

A large data set study of 135 hospitals in the United Kingdom from 2010 to 2017 confirmed that antibiotic usage did not correlate with patient morbidity and mortality [21]. Other studies in neonates have shown that the use of broad-spectrum antibiotics within 1 week after birth increased the risks of LOS, NEC, and death [22]. Letouzey et al. [23] found that early postnatal empirical antibiotic therapy in preterm infants at low risk of EOS was associated with a higher risk of severe cerebral lesions and moderate to severe BPD. In addition, unnecessary antibiotic use in the early postnatal period may delay the time required to reach full enteral nutrition. Saleem et al. [24] have demonstrated that the longer the duration of antibiotic use in preterm infants, the greater the delay in their feeding. Many scholars now believe that early antibiotic use in VLBW infants can lead to intestinal flora dysbiosis [25,26], which increases the risk of adverse outcomes such as NEC [27], prolongs the length of hospitalization, and increases the medical costs in preterm infants. In this study, the time to reach full enteral nutrition was also reduced significantly over the initial 3 years of the study period, suggesting that shortening the duration of early empirical antibiotic therapy may facilitate the establishment of normal intestinal flora and therefore decrease the time to reach full enteral feeds, and subsequently the incidence of LOS.

As a retrospective study, it does have inherent limitations. The first limitation is that the main data are more than 6 years old, even though this study employed recent data from 2021 to demonstrate the continued influence of this antibiotic usage strategy. Due to possible practice changes over the last 5 years, we did not include the data from 2021 for logistic regression analysis. Secondly, the reduction in antibiotic abuse may decrease the emergence of MDR bacterial pathogens in the NICU. Unfortunately, the authors do not have data about the effects of antibiotic practice changes in VLBW infants on the prevalence of MDR in this NICU. Another limitation of this study is that as a single-center retrospective cohort study with a time span of >3 years, it is possible that the effect of daily practice changes is not controlled through multivariate regression analysis. A multi-center clinical trial is needed to confirm the effects and the safety of restricting antibiotic use in VLBW infants. Nevertheless, this study is one of the first aimed at improving the antibiotic use strategy for VLBW infants in China, with notable improvements in terms of antibiotic practice. We hope that this study will encourage more related studies, in an effort to improve the prognosis of VLBW infants in China.

## 4. Methods

### 4.1. Study Subjects and Inclusion Criteria

This is a single-center retrospective cohort study performed in the NICU of Xiangya Hospital of Central South University. The authors initiated an antibiotic stewardship program in early 2015. In order to examine the longitudinal effect of this stewardship program, all eligible VLBW infants from 2014 to 2016 were enrolled into the study and divided the study population into three groups based on their delivery dates: in 2014, 2015, and 2016. For analysis, the year 2014 was classified as pre-stewardship, 2015 as during stewardship, and 2016 as post-stewardship.

From 1 January 2014 to 31 December 2016, the VLBW infants admitted to the NICU were enrolled and the following data were collected from the medical records: (1) gestational age, birth weight, sex, and relevant resuscitation in the delivery room; (2) relevant maternal perinatal data; (3) initial NICU respiratory management; (4) duration of antibiotic use (initial postnatal antibiotic course and total antibiotic use); (5) days to full enteral feeding, and duration of hospitalization; (6) major short-term outcomes including death, late onset sepsis (LOS), necrotizing enterocolitis (NEC), severe intraventricular hemorrhage (IVH), severe bronchopulmonary dysplasia (BPD), and ≥3 stage retinopathy of prematurity (ROP).

The exclusion criteria include (1) those admitted after 24 h of birth; (2) those with culture proven early onset sepsis (EOS); (3) those who died within 1 week of life; (4) those with major malformations; (5) those with suspected or confirmed genetic metabolic diseases; (6) those with incomplete medical records. During the three years of the study, there were no major changes in the respiratory support equipment and strategies, NICU physical environment, or staffing conditions. The routine testing of various specimens was performed by the clinical laboratory of Xiangya Hospital of Central South University.

To assess the continuity and impacts of antibiotic stewardship interventions, data from 2021 were also employed and compared to 2016. The same inclusion and exclusion standards were applied, and similar data were collected. In 2021, a total of 105 VLBW infants were recruited, and after excluding 12 cases hospitalized after 24 h of birth and 3 cases with suspected or confirmed genetic metabolic disorders, 90 VLBW infants were eventually enrolled.

The study was approved by the Ethics Committee of Xiangya Hospital of Central South University and conducted in full compliance with medical ethics standards. Clinical data were collected retrospectively from the hospital medical records. A waiver for parental informed consent had been granted from the Institutional Ethics Committee (ethical approval number: 202112258).

### 4.2. Relevant Definitions

The definition of sepsis was based on the most recent criteria outlined by the China National Committee [13], and has been described in detail in the authors’ previous expert opinion piece [10]. Briefly, it refers to a neonate with a positive blood or cerebrospinal fluid culture. If the blood culture is negative for bacterial pathogens, sepsis is also diagnosed when the patient has a deterioration in medical condition (e.g., with poor response, irritability, respiratory distress, temperature instability, poor perfusion, etc.) along with ≥2 abnormal blood nonspecific infection indicators (white blood cell count, platelet count, ratio of immature to total neutrophil counts, CRP, and PCT), or with changes in the cerebrospinal fluid examination consistent with meningitis. PROM refers to the rupture of membranes >18 h. NEC refers to patients who meet the revised Bell diagnostic criteria stage 2 or above [28]. Severe BPD means meeting the National Institute of Child Health and Human Development (NICHD) 2018 revised diagnostic criteria for severe BPD [29]. Severe IVH refers to those with grade III or IV intraventricular hemorrhage on cranial ultrasound or brain MRI [30]. Early antibiotic course refers to days on antibiotics from birth to the end of the first course of antibiotics. Total antibiotic usage refers to the total days of use of intravenous antibiotics during the entire NICU hospitalization, which is expressed as antibiotic use rate (AUR), i.e., the ratio of the total days on antibiotics to the entire hospitalization days during the NICU stay [31,32].

### 4.3. Statistical Analysis

Statistical analysis was performed using SPSS 22.0 statistical software (SPSS, Inc., Chicago, IL, USA). Normally distributed data were expressed as mean and standard ($\overset{-}{X}$ ± S) and non-normally distributed variables were expressed by median and interquartile range (25–75 percentile). One-way ANOVA was used for normally distributed variables, and for comparisons between the two groups, the ANOVA method of multiple comparisons was used. Additionally, a *t*-test was used for normally distributed variables between two groups. The Kruskal-–Wallis H test was used for non-normally distributed variables, and comparisons between groups used the mean rank’s post hoc multiple comparison of Kruskal–Wallis H test. The categorical data were expressed as rates (%) and the Chi-square test or Fisher’s exact probability method, and the Chi-square splitting method were used for comparisons between groups. For comparisons across groups, the test level was modified to a statistically significant difference of *p* < 0.0167. Univariable and multivariable logistic regressions were used to evaluate the associations between antibiotic use and the composite clinical short-term adverse outcome (with ≥III degree IVH, LOS, ≥2 stage NEC, severe BPD, ≥3 stage ROP, or death, one of which is assigned a value of 1, and without ≥III degree IVH, LOS, ≥2 stage NEC, severe BPD, ≥3 stage ROP, or death, any one of which is assigned a value of 0) as the dependent variable, and grouping variables, early antibiotic course distribution and AUR as independent variables, correcting for confounding factors such as gestational age, birth weight, Apgar scores, etc. Two-sided *p* < 0.05 was considered as representing statistical significance.

## 5. Conclusions

With the introduction of an antibiotic stewardship program which includes formulating specific guidelines and provider education for the diagnosis and treatment of sepsis in VLBW infants, the duration of empirical antibiotic use in VLBW infants was significantly reduced. The reduction in antibiotic usage was associated with a decreased odds of having an adverse composite short-term outcome, suggesting that restricting antibiotic use in VLBW infants is beneficial and can be achieved safely and effectively in China. The study also demonstrates that the antibiotic stewardship program in a NICU is sustainable.

## Figures and Tables

**Figure 1 antibiotics-12-00741-f001:**
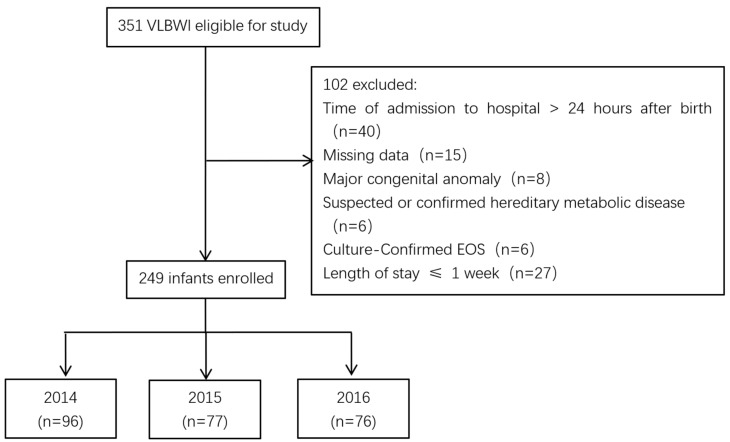
Flow diagram of the enrolled patients.

**Table 1 antibiotics-12-00741-t001:** Baseline perinatal characteristics of the 3 groups.

Variable	2014 (*n* = 96)	2015 (*n* = 77)	2016 (*n* = 76)		*p*
**Neonatal characteristics**					
GA, mean (SD)	30.3 ± 1.9	30.0 ± 1.9	29.8 ± 2.1	*F* = 1.787	0.170
BW, mean (SD)	1.23 ± 0.17	1.24 ± 0.19	1.17 ± 0.19	*F* = 2.963	0.054
SGA, *n* (%)	29 (30.2)	16 (20.8)	21 (27.6)	*χ*^2^ = 2.021	0.364
Male, *n* (%)	50 (52.1)	43 (55.8)	51 (67.1)	*χ*^2^ = 4.106	0.128
PROM, *n* (%)	31 (32.3)	27 (35.1)	21 (27.6)	*χ*^2^ = 0.999	0.607
Apgar score ≤ 7 at 1 min, *n* (%)	36 (37.5)	23 (29.9)	38 (50.0)	*χ*^2^ = 9.330	0.053
Apgar score ≤ 7 at 5 min, *n* (%)	4 (4.2)	5 (6.5)	10 (13.2) *	*χ*^2^ = 9.907	0.042
Endotracheal intubation in the delivery room, *n* (%)	11 (11.5)	9 (11.7)	8 (10.5)	*χ*^2^ = 0.059	0.971
Surfactant usage, *n* (%)	60 (62.5)	56 (72.7)	56 (73.7)	*χ*^2^ = 3.180	0.204
Mechanical ventilation for first 3 days after birth, *n* (%)	12 (12.5)	10 (13.0)	13 (17.1)	*χ*^2^ = 0.850	0.654
**Maternal characteristics**					
Maternal age (y)	30.0 (26.0, 33.0)	30.0 (27.0, 33.0)	30.5 (27.0, 34.0)	*Z =* 0.617	0.734
Cesarean delivery, *n* (%)	67 (69.8)	48 (62.3)	45 (59.2)	*χ*^2^ = 2.247	0.325
Multiples, *n* (%)	36 (37.5)	36 (46.8)	35 (46.1)	*χ*^2^ = 1.917	0.384
IVF, *n* (%)	30 (31.3)	29 (37.7)	24 (31.6)	*χ*^2^ = 0.942	0.624
Antepartum antibiotics, *n* (%)	40 (42.1)	23 (29.9)	37 (48.7)	*χ*^2^ = 5.830	0.054
Antenatal steroids, *n* (%)	66 (68.8)	48 (62.3)	57 (75.0)	*χ*^2^ = 2.851	0.240

* Compared with the 2014 group: *p* < 0.0167; GA: Gestational age; BW: birth weight; SGA: small for gestational age; PROM: premature rupture of membrane > 18 h; IVF: in vitro fertilization.

**Table 2 antibiotics-12-00741-t002:** Postnatal empirical antibiotic usage and short-term outcomes among the 3 groups.

Empirical Antibiotics	2014 (*n* = 96)	2015 (*n* = 77)	2016 (*n* = 76)		*p*
Early antibiotic usage rate, *n* (%)	95 (99.0)	72 (93.5)	70 (92.1)	*χ*^2^ = 5.024	0.081
Early antibiotic usage, days, *M*(*Q*_1_, *Q*_3_)	25.0 (17.0, 46.0)	13.5 (8.0, 20.0) *	5.0 (3.0, 12.0) *^#^	*Z* = 86.117	<0.001
**Early antibiotic course** **distribution**					
≤3 d	2 (2.1)	7 (9.1) *	29 (38.2) *^#^	*χ*^2^ = 73.325	<0.001
4~7 d	2 (2.1)	9 (11.7) *	17 (22.4) *^#^		
>7 d	92 (95.8)	61 (79.2) *	30 (39.5) *^#^		
Total antibiotic usage, days, *M*(*Q*_1_, *Q*_3_)	27.0 (18.0, 50.3)	21.0 (12.5, 35.0) *	10.0 (4.0, 20.0) *^#^	*Z* = 50.654	<0.001
Antibiotics use rate	0.64 (0.42, 0.89)	0.39 (0.17, 0.53) *	0.12 (0.07, 0.25) *^#^	*Z* = 99.289	<0.001
**Outcomes of infants**					
Death, *n* (%)	3 (3.1)	7 (9.1)	3 (3.9)	*χ*^2^ = 3.065	0.242
NEC, *n* (%)	7 (7.3)	8 (10.4)	10 (13.2)	*χ*^2^ = 1.631	0.442
Severe IVH, *n* (%)	24 (25.0)	16 (20.8)	6 (7.9) *	*χ*^2^ = 8.634	0.013
LOS, *n* (%)	23 (24.0)	18 (23.4)	9 (11.8)	*χ*^2^ = 4.635	0.099
Severe ROP, *n* (%)	10 (10.4)	10 (13.0)	8 (10.5)	*χ*^2^ = 0.339	0.844
Severe BPD, *n* (%)	5 (5.2)	5 (6.5)	6 (7.9)	*χ*^2^ = 0.595	0.724
Days to full enteral feeding (d)	26.0 (20.0, 36.5) (*n* = 93)	21.5 (14.8, 29.5) (*n* = 70) *	17.0 (12.0, 24.0) (*n* = 73) *^#^	*Z* = 26.203	<0.001
Length of hospitalization (d)	43.0 (30.5, 68.0) (*n* = 93)	40.0 (28.8, 55.3) (*n* = 70)	44.0 (30.5, 59.5) (*n* = 73)	*Z* = 1.647	0.439

* Compared with the 2014 group: *p* < 0.0167; ^#^ compared with the 2015 group: *p* < 0.0167; antibiotics use rate: ratio of days of antibiotic use to days of hospitalization.

**Table 3 antibiotics-12-00741-t003:** Logistic Regression Analysis.

Exposure	Composite Adverse Outcomes		Composite Adverse Outcomes	
	OR (95%CI)	*p*	aOR (95%CI)	*p*
Year of antibiotic use				
2014	1.0		1.0	
2015	1.061 (0.579, 1.942)	0.849	0.865 (0.387, 1.933)	0.723
2016	0.512 (0.269, 0.976)	0.042	0.149 (0.055, 0.401)	<0.001
Initial antibiotic course				
0~3 d	1.0		1.0	
4~7 d	2.640 (0.758, 9.194)	0.127	4.407 (0.973, 19.968)	0.054
>7 d	5.478 (2.047, 14.663)	0.001	5.148 (1.598, 16.583)	0.006
AUR (every 10%)	1.181 (1.084, 1.288)	<0.001	1.175 (1.047, 1.319)	0.006

Composite adverse outcomes: death, NEC, severe IVH, LOS, severe ROP, or severe BPD. aOR: adjusted for male, GA, BW, IVF, endotracheal intubation in the delivery room, cesarean delivery, PROM, antenatal steroids, mechanical ventilation for first 3 days after birth, surfactant usage, multiple, Apgar score ≤ 7 at 5 min, and maternal age.

**Table 4 antibiotics-12-00741-t004:** Baseline perinatal characteristics of the 2016 and 2021 groups.

Variable	2016 (*n* = 76)	2021 (*n* = 90)		*p*
**Neonatal characteristics**				
GA, mean (SD)	29.8 ± 2.1	30.5 ± 2.1	*t* = −2.413	0.017
BW, mean (SD)	1.17 ± 0.19	1.26 ± 0.15	*t* = −3.148	0.002
SGA, *n* (%)	21 (27.6)	22 (24.4)	*χ*^2^ = 0.218	0.641
Male, *n* (%)	51 (67.1)	43 (47.8)	*χ*^2^ = 6.267	0.012
PROM, *n* (%)	21 (27.6)	20 (22.2)	*χ*^2^ = 0.648	0.421
Apgar score ≤ 7 at 1 min, *n* (%)	38 (50.0)	36 (40.0)	*χ*^2^ = 3.071	0.215
Apgar score ≤ 7 at 5 min, *n* (%)	10 (13.2)	7 (7.8)	*χ*^2^ = 3.832	0.147
Endotracheal intubation in the delivery room, *n* (%)	8 (10.5)	10 (11.1)	*χ*^2^ = 0.015	0.904
Surfactant usage, *n* (%)	56 (73.7)	49 (54.4)	*χ*^2^ = 6.562	0.010
Mechanical ventilation for first 3 days after birth, *n* (%)	13 (17.1)	12 (13.3)	*χ*^2^ = 0.458	0.498
**Maternal characteristics**				
Maternal age (y)	30.5 (27.0, 34.0)	32.0 (29.0, 34.0)	*Z =* −1.025	0.305
Cesarean delivery, *n* (%)	45 (59.2)	76 (84.4)	*χ*^2^ = 13.278	<0.001
Multiples, *n* (%)	35 (46.1)	40 (44.4)	*χ*^2^ = 0.043	0.836
IVF, *n* (%)	24 (31.6)	36 (40.0)	*χ*^2^ = 1.266	0.261
Antepartum antibiotics, *n* (%)	37 (48.7)	38 (42.2)	*χ*^2^ = 0.695	0.405
Antenatal steroids, *n* (%)	57 (75.0)	78 (86.7)	*χ*^2^ = 3.693	0.055

**Table 5 antibiotics-12-00741-t005:** Postnatal empirical antibiotic usage and short-term outcomes between the 2016 and 2021 groups.

Empirical Antibiotics	2016 (*n* = 76)	2021 (*n* = 90)		*p*
Early antibiotic usage rate, *n* (%)	70 (92.1)	60 (66.7)	*χ*^2^ = 15.700	<0.001
Early antibiotic usage, days, *M*(*Q*_1_, *Q_3_*)	5.0 (3.0, 12.0)	4.0 (3.0, 7.0)	*Z* = −3.703	<0.001
**Early antibiotic course** **distribution**				
≤3 d	29 (38.2)	51 (56.7)	*χ*^2^ = 7.636	0.022
4~7 d	17 (22.4)	20 (22.2)		
>7 d	30 (39.5)	19 (21.1)		
Total antibiotic usage, days, *M*(*Q*_1_, *Q*_3_)	10.0 (4.0, 20.0)	7.0 (4.0, 16.0)	*Z* = −2.592	0.010
Antibiotics use rate	0.12 (0.07, 0.25)	0.07 (0.00, 0.12)	*Z* = −4.446	<0.001
**Outcomes of infants**				
Death, *n* (%)	3 (3.9)	2 (2.2)	*χ*^2^ = 0.420	0.661
NEC, *n* (%)	10 (13.2)	8 (8.9)	*χ*^2^ = 0.777	0.378
Severe IVH, *n* (%)	6 (7.9)	4 (4.4)	*χ*^2^ = 0.866	0.515
LOS, *n* (%)	9 (11.8)	10 (11.1)	*χ*^2^ = 0.022	0.883
Severe ROP, *n* (%)	8 (10.5)	11 (12.2)	*χ*^2^ = 0.117	0.732
Severe BPD, *n* (%)	6 (7.9)	4 (4.4)	*χ*^2^ = 0.866	0.352
Days to full enteral feeding (d)	17.0 (12.0, 24.0) (*n* = 73)	24.0 (17.0, 36.0) (*n* = 88)	*Z* = −4.057	<0.001
Length of hospitalization (d)	44.0 (30.5, 59.5) (*n* = 73)	44.0 (34.0, 60.0) (*n* = 88)	*Z* = −0.160	0.873

## Data Availability

The data are available on request.

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
