# Peer review of "Restrictive Use of Empirical Antibiotics Is Associated with Improved Short Term Outcomes in Very Low Birth Weight Infants: A Single Center, Retrospective Cohort Study from China"

_antibiotics, 2023, doi:10.3390/antibiotics12040741_

Round 1

Reviewer 1 Report

This is a very interesting study, the overall study design and results presentation is up to the mark, however, I am adding two suggestions for methodology, which might further improve the contents.

1. can you please specify the antibiotics class/types? it will be more specific

 2. What about the scenario of antibiotic resistance in the NICU? you might be able to get that data from the hospital surveillance system. 

Author Response

We would like to express our sincere gratitude to all the reviewers for their positive feedback and constructive comments, which are quite helpful to improve the quality of this manuscript. We have tried our best to address all these comments and revised the manuscript as suggested. Here we provide our responses to the reviewers’ comments item by item.

Point 1: can you please specify the antibiotics class/types? it will be more specific

Response 1: The first line antibiotics used for empirical treatment of suspected EOS in our unit were Ampicillin/Sulbactam, Amoxicillin/Clavulanic acid, or ampicillin plus one third generation of cephalosporin. We agree with the reviewer that the importance to specify the antibiotics class/types. We have added this sentence into our manuscript.

Point 2: What about the scenario of antibiotic resistance in the NICU? you might be able to get that data from the hospital surveillance system. 

Response 2:The reduction of antibiotic abuse may decrease the emergence of MDR bacterial pathogens in the NICU. Unfortunately, we do not have data about the effects of antibiotic practice changes in VLBW infants on the prevalence of MDR in this NICU, and this is a limitation of the current study. This limitation has been discussed in the manuscript.

Reviewer 2 Report

This is an interesting retrospective cohort study of changes in antibiotic use through implementation of antimicrobial stewardship between 2014 and 2016, then 2016 compared to 2021. The following comments are offered in order to strengthen the presentation of findings and manuscript.

Title: needs to include that this was a single center, retrospective cohort study.

Abstract:  Consider changing "side effects" to "adverse effects" in the abstract and throughout the manuscript. It's a more accurate and descriptive term conveying risk.

In the abstract, your argument would be strengthened with inclusion of statistics (odds ratios, confidence intervals, and p-values). Adding p-values would increase the likelihood that someone would read your manuscript.

"Over the 3 years period, the duration of an initial 24 antibiotic course was significantly reduced. The proportion of patients receiving an initial antibiotic course for ≤3 days gradually increased, while the proportion of babies treated with an initial antibiotic course >7 days significantly decreased." (odds ratio, p-value?)

"Total days of antibiotic usage during the entire NICU stay also showed a significant reduction. (%, p-value)?

"After adjusting for confounders, the reduction in antibiotic usage was associated with decreased odds of having an adverse composite short-term outcome defined as the presence of any of the listed conditions (death, NEC, LOS, severe IVH, severe ROP, or severe BPD)." Too many to list separately; however, you could combine them and report odds ratio as a composite.

"To assess the continuity of antibiotic stewardship in the NICU, data from 2021 were also analyzed and compared to 2016. The median duration of an initial antibiotic course further decreased from 5.0 days in 2016 to 4.0 days in 2021 (p-value?)."

Keywords: inclusive

Introduction covers recent and appropriate literature. Ln 75: include a definition of VLBW with a citation.

Results: Ln 84: you are implying that patients were "subjects" that enrolled and participated in a clinical trial. You need to explain how the patients were identified from healthcare records. Further, you should explain that the review criteria were generated prior to commencement of the data collection and analysis.

There is no mention of the medications that patients received. Prophylaxis with the indicated antibiotics should be included in the write-up.

Limitations are well explained and justified.

Methods: IRB approval however the study was not registered at any research repository.

References: minor formatting changes are needed to comply with MDPI style.

Well done! Thank you for the opportunity to review and comment on your important study.

Author Response

We would like to express our sincere gratitude to all the reviewers for their positive feedback and constructive comments, which are quite helpful to improve the quality of this manuscript. We have tried our best to address all these comments and revised the manuscript as suggested. Here we provide our responses to the reviewers’ comments item by item.

Point 1: Title: needs to include that this was a single center, retrospective cohort study.

Response 1: Thank you for your useful suggestion, we have added relevant content in the title section.

Point 2: Abstract:  Consider changing "side effects" to "adverse effects" in the abstract and throughout the manuscript. It's a more accurate and descriptive term conveying risk.

Response 2: Thank you for your suggestion, we have made corresponding modifications throughout the manuscript.

Point 3: In the abstract, your argument would be strengthened with inclusion of statistics (odds ratios, confidence intervals, and p-values). Adding p-values would increase the likelihood that someone would read your manuscript.

Response 3: Thank you for your useful suggestion, we have added relevant content in the abstract.

 Point 4: Introduction covers recent and appropriate literature. Ln 75: include a definition of VLBW with a citation.

Response 4: Thank you for your useful suggestion, we have added relevant content in the section.

Point 5: Results: Ln 84: you are implying that patients were "subjects" that enrolled and participated in a clinical trial. You need to explain how the patients were identified from healthcare records. Further, you should explain that the review criteria were generated prior to commencement of the data collection and analysis.

Response 5: The study was a single center, retrospective cohort study, and we collected clinical data from retrospective chart review. The inclusion criteria are detailed in the method section.  

Point 6: There is no mention of the medications that patients received. Prophylaxis with the indicated antibiotics should be included in the write-up.

Response 6: The first line antibiotics used for empirical treatment of suspected EOS in our unit were Ampicillin/Sulbactam, Amoxicillin/Clavulanic acid, or ampicillin plus one third generation of cephalosporin. We have added this sentence into our manuscript.

Point 7: Methods: IRB approval however the study was not registered at any research repository.

Response 7: The study was not considered to be a clinical trial when we initiated the antibiotic stewardship program in 2015. Therefore, we did not register it at that time in any of relevant research repository website.

Point 8: References: minor formatting changes are needed to comply with MDPI style.

Response 8:  Thank you for your useful suggestion, we have made corresponding modifications in this section. 

Reviewer 3 Report

The article is a retrospective analysis of the antibiotic treatment introduced for babies born with very low birth weight (VLBW) after introducing antibiotic stewardship in the neonatal intensive care unit (NICU). They concluded that reducing the length of the antibiotic prophylaxis had not been followed with any adverse effects, or unfavorable outcomes, instead, some positive observations were recorded as suggesting that restricting antibiotic use in 356 VLBW infants is beneficial and can be achieved safely and effectively.

The abbreviations of different expressions should stand after the first mention of them. In the line 30-31 the abbreviations (NEC, LOS, severe IVH, severe ROP, or 30 severe BPD) the full expression of the phrases must be clearly included before the abbreviations. Considering the large number of abbreviations, I propose a glossary included after the abstract.  

Author Response

We would like to express our sincere gratitude to all the reviewers for their positive feedback and constructive comments, which are quite helpful to improve the quality of this manuscript. We have tried our best to address all these comments and revised the manuscript as suggested. Here we provide our responses to the reviewers’ comments item by item.

Point 1: The abbreviations of different expressions should stand after the first mention of them. In the line 30-31 the abbreviations (NEC, LOS, severe IVH, severe ROP, or 30 severe BPD) the full expression of the phrases must be clearly included before the abbreviations. Considering the large number of abbreviations, I propose a glossary included after the abstract. 

Response 1: Thank you for your useful suggestion, we have made corresponding modifications in this section.